# Successful Treatment of Renal Cell Carcinoma Associated with Hypertrophic Osteopathy in a Cat

**DOI:** 10.3390/vetsci11120669

**Published:** 2024-12-20

**Authors:** Takashi Tanaka, Midori Tanaka, Tomoyuki Tezuka, Kazumi Shimada, Ryo Tanaka

**Affiliations:** 1Department of Veterinary Science, Tokyo University of Agriculture and Technology, Tokyo 183-8509, Japan; kazumi-s@go.tuat.ac.jp (K.S.); fu0253@go.tuat.ac.jp (R.T.); 2Takashimadaira Tezuka Animal Hospital, Tokyo 175-0082, Japan; mabari310@gmail.com (M.T.); ttahdvm@gmail.com (T.T.)

**Keywords:** hypertrophic osteopathy, renal cell carcinoma, nephrectomy

## Abstract

Reports of primary feline renal tumors are rare and generally limited to case reports and surveys of tumors, of which renal cell carcinoma is the most common, excluding lymphoma. Renal cell carcinoma is often a clinically silent disease. Consequently, it is generally diagnosed at a late stage. It has a moderate metastatic rate at the time of diagnosis and a high metastatic rate at an advanced stage. Hypertrophic osteopathy in cats is rare, so there are few reports in the literature. Hypertrophic osteopathy is generally secondary to chronic pulmonary infiltrative lesions. This article presents a rare case of renal tumor associated with hypertrophic osteopathy, in which the patient experienced long-term survival after the surgical removal of the renal tumor, as well as a resolution of the signs associated with the bone lesions.

## 1. Introduction

Reports of primary feline renal tumors are rare and generally limited to case reports and surveys of tumors. The most common renal tumor in cats is lymphoma, and the next is renal carcinoma. Renal lymphoma is almost always bilateral [1], so a unilateral renal tumor in a cat strongly suggests carcinoma [2]. Hypertrophic osteopathy (HO) is most often observed in dogs, horses, cattle, wild animals, and humans [3]. In these species, HO is generally secondary to chronic pulmonary infiltrative lesions, which are metastatic or primary neoplasms in the lungs that are labelled HO of pulmonary origin [4]. Similarly, HO in cats is most commonly associated with intrathoracic tumors, although rare cases have been reported in association with intraperitoneal mass lesions [5,6]. Here, the present article reports a case of a cat with renal cell carcinoma presenting with HO, in which a long-term survival was achieved following surgical treatment.

## 2. Case Description

An eight-year-old spayed female Abyssinian cat (weighting 3.75 kg) presented because of lameness for several months, reluctance to flex her hind limbs or climb to high places. Palpation revealed swelling, heat, and a reduced range of motion in the tarsal joints on both sides. A neurological examination revealed no abnormalities. A radiographic examination of both hind limbs revealed soft tissue swelling the distal end of the hind limbs and palisade-like periosteal proliferation from the distal tibia to the talus and metatarsal on both sides (Figure 1).

The examination of thoracic radiographs showed no abnormal findings (Figure 2A,B). The examination of abdominal radiographs showed a mass in the area of the left kidney (Figure 2C,D). No abnormal findings were observed in the complete blood count and routine serum biochemistry tests. An abdominal ultrasound revealed a hypoechoic renal mass located in the caudal pole of the left kidney (Figure 2E). The mass was 27 mm in the longest dimension and exhibited a clear border with the surrounding tissue. The right kidney was normal, and no abnormalities were detected in the other abdominal organs.

For the purpose of diagnosing the left renal mass, fine needle aspiration was performed on the mass lesion under ultrasound guidance, and irregularly round glandular epithelial cells that differed from those in the kidney tissue were observed. These cells had a high N/C ratio and irregularly round atypical nuclei with varying sizes of nucleoli. Based on these findings, the cat was strongly suspected of having a left renal tumor associated with HO. A nephrectomy of the left renal tumor was performed. Premedication was administered with fentanyl (5 μg/kg IV). General anesthesia was induced with propofol (7 mg/kg IV) and maintained with isoflurane administered in 100% oxygen. Intraoperative constant rate infusion of fentanyl (5 μg/kg/h) was administered for analgesia. The abdomen was approached through a midline celiotomy (Figure 3).

Exploration of the abdominal cavity indicated a small amount of clear serous fluid accumulated between the fat covering the left kidney and the retroperitoneum. The retroperitoneum was incised to expose the left kidney, and a voluminous mass deforming the left kidney was confirmed. There was adhesion between the caudal region of the left kidney and the fatty tissue in the retroperitoneum; however, no tumor infiltration to other areas or lymph node enlargement was evident. After a thorough visual inspection of the abdominal cavity, the left kidney was removed using a standard procedure. A histopathological examination revealed renal cell carcinoma. The tumor cells were irregularly round glandular epithelial cells and formed a solid glandular tubular structure. They were formed protruding outward from the renal cortex. There was a slight variation in size among the tumor cells and a small number of mitotic figures were observed in the irregularly round atypical nuclei. A renal capsule structure was observed at the edge of the tumor nest. However, the membrane-like structure was unclear in some areas and continuous with adipose tissue. No obvious vascular invasion was observed (Figure 4).

The cat recovered uneventfully from the anesthesia and surgery and was discharged after five days. Upon follow-up one month after the surgery, a radiographic examination of both hind limbs revealed the periosteal proliferation of the distal tibia on both sides had decreased. The lameness had improved and reduced swelling in the tarsal joints on both sides. A radiographic exam two months later revealed that the periosteal hyperplasia at the distal tibia had completely regressed, and the soft tissue swelling around the tarsal joint had decreased along with the periosteal proliferation around the talus; however, periosteal proliferation did not completely disappear in the bones distal to the talus (Figure 5). No abnormal findings were observed in the blood tests. A renal prescription diet was started in consideration of the remaining right kidney at this point.

Six months after surgery, the radiographic examinations showed that these lesions had not decreased, but there were no clinical signs such as lameness, therefore, follow-up radiographic examinations of both hind limbs were not performed.

One year after surgery, urine analysis, blood biochemistry, thorax and abdominal radiography, and abdominal ultrasound showed no abnormalities. At 1670 days following surgery, the cat was presented to our hospital with polyuria and polydipsia. Blood tests indicated that urea nitrogen and creatinine were 33.4 mg/dL (reference range: 17.6–32.8 mg/dL) and 2.1 mg/dL (reference range: 0.9–2.0 mg/dL), respectively. Urinalysis revealed a decrease in urine specific gravity. An abdominal ultrasound showed no abnormalities in the right kidney. The cat was suspected of having chronic kidney disease (CKD) because of continued azotemia and low specific gravity urine (1.015). The cat’s last blood test performed eight years after surgery, showed that urea nitrogen and creatinine were 40.7 mg/dL and 2.4 mg/dL, respectively. Anemia or proteinuria was not noted at the time. The cat was subsequently fed on a renal prescription diet but received no additional examinations and treatments. The cat’s activity and appetite gradually decreased and died 4645 days following the operation. The cause of death could not be determined as no autopsy was performed.

## 3. Discussion

HO is a syndrome characterized by periosteal hyperplasia along the long axis of the long bones that occurs in association with malignant tumors or other diseases and often occurs in association with primary lung tumors [7]. Clinical signs include limb swelling and pain, and a diagnosis is made by radiography of the affected limb showing a periosteal reaction. In this case, the cat also presented with lameness and pain. Because an excision biopsy was not performed on the periosteal lesion, bone tumors and metastatic lesions could not be ruled out. However, after the removal of the renal tumor, periosteal proliferation in both hind limbs decreased, suggesting that the cat had developed secondary HO. Several hypotheses have been used to explain the pathogenesis of secondary HO. Humoral [8] and neuronal theories [6] have been postulated to explain the increase in blood flow to the limbs; however, the pathogenesis of HO remains unclear as there have been reports of cats developing idiopathic HO [9]. The prognosis for secondary HO in cats varies, ranging from regression 15 weeks following surgical resection of an adrenocortical carcinoma, to 6 months after removal of a pulmonary sarcoma, to no radiographic changes 9 months after renal adenocarcinoma removal [10,11,12]. Two months after the renal cell carcinoma was removed, the periosteal hyperplasia at the distal tibia had completely regressed and the periosteal hyperplasia around the talus was decreased; however, there were no changes in subsequent radiographic analyses. Therefore, it is difficult to conclude as to whether the postoperative degree of reduction in periosteal proliferation may be used as a prognostic factor.

Primary renal neoplasia is not frequently observed in feline medicine, with an estimated prevalence of less than 1% of all cancers [12]. Other than lymphoma, the majority of tumors are epithelial, with renal cell carcinoma considered the most common, followed by transitional cell carcinoma, squamous cell carcinoma, and renal adenoma [5,13,14,15]. Renal leiomyosarcoma, myxoma, myxosarcoma, and paraganglioma have also been reported [16,17,18,19]. As most tumors are unilateral, nephrectomy may represent a treatment option, provided no metastasis has occurred. The role of adjunctive therapy, such as chemotherapy or immunotherapy, has not yet been established. As with most primary renal cancers, renal cell carcinoma is often a clinically silent disease. Consequently, it is generally diagnosed at a late stage. It has a moderate metastatic rate (16–34%) at the time of diagnosis and a high metastatic rate (70–75%) at an advanced stage [20,21]. Regardless of the tumor type, the successful management of primary renal neoplasia relies on the complete excision of the mass. Survival time after renal cell carcinoma excision in cats varies (28–2292 days), with some deaths occurring during the perioperative period [22]. Therefore, how quickly and effectively the kidney tumor can be removed has a major impact on survival. Moreover, routine health exams are important in middle-aged to elderly cats because kidney tumors rarely exhibit clinical signs. Because the cat in the present study had developed secondary HO to renal cell carcinoma, this represented a rare case in which nephrectomy could be performed before the clinical signs of renal cell carcinoma were evident. If the signs of hypertrophic osteopathy are observed in a cat, it is important to check for tumors not only in the thoracic cavity but also in the abdominal cavity. Conversely, no vascular invasion was observed at the time of nephrectomy; however, adhesions were observed between the caudal side of the left kidney and the fat in the retroperitoneum. Renal cell carcinomas are remarkably resistant to chemotherapy, and the development of drug resistance is common in humans [23]. The expression of the multidrug-resistant transporter in tumor cells is one mechanism associated with drug resistance in renal cell carcinoma [24]. In addition, conventional cytotoxic agents, such as carboplatin, doxorubicin, cyclophosphamide, cisplatin, palladia, and mitoxantrone, were administered to dogs after nephrectomy, but no statistical difference in survival was observed compared with the untreated animals [21]. Regardless of tumor type, primary renal tumors have a high risk of metastasis, and some form of adjuvant chemotherapy is usually recommended; however, because of the lack of success of chemotherapy in dogs and humans, the focus of successful treatment remains complete tumor resection [25]. In the present study, no additional chemotherapy was administered to the cat; however, early surgical excision of the renal cell carcinoma enabled it to survive for more than 4500 days without additional treatment.

There were two limitations of the management to this case report. First, there was a possibility that the kidney prescription diet was started too early. Dietary manipulation is a mainstay of CKD. A renal prescription diet was started in consideration of the remaining right kidney, there was little evidence that this alone has a major effect on the progression of CKD [26]. Second, there was a lack of routine evaluation of CKD. When CKD is suspected, a minimum routine database should ideally include routine urinalysis and serum biochemistry, systolic blood pressure and diagnostic imaging. The blood tests and urinalysis were performed monthly after azotemia was confirmed in this case, but measurement of thyroxine or urine culture was not performed. The final blood test was performed 8 years after surgery, but subsequent tests were not performed due to the owner’s request. Therefore, details regarding renal function assessment and concomitant diseases in the end stage were unknown.

## 4. Conclusions

There have been few reports of feline HO in the literature and only a few have been associated with abdominal tumors. To our knowledge, this is the first case of feline renal cell carcinoma associated with HO, in which a long-term prognosis was achieved with surgical treatment alone.

## Figures and Tables

**Figure 1 vetsci-11-00669-f001:**
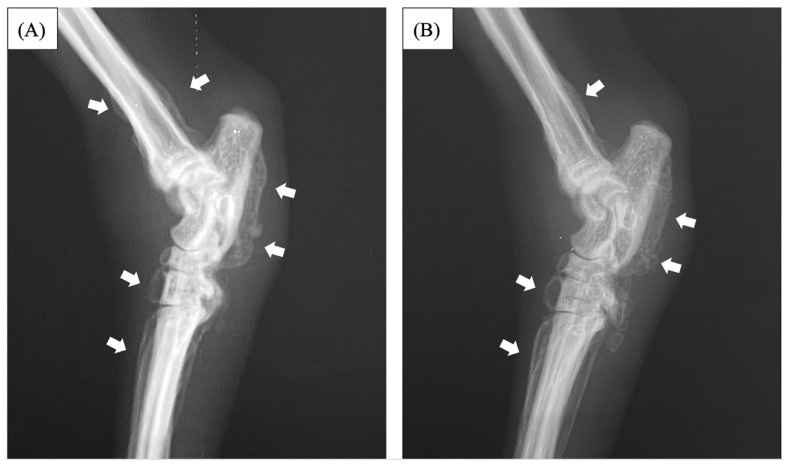
Radiographs images of a lateral view of the distal ends of the right and left hind limbs. Radiographic examination of the right hind limb (**A**) and left hind limb (**B**) showed palisade-like periosteal proliferation in the distal tibia, talus, and metatarsal (thick white arrow). No abnormalities were observed on the articular surfaces of either limb.

**Figure 2 vetsci-11-00669-f002:**
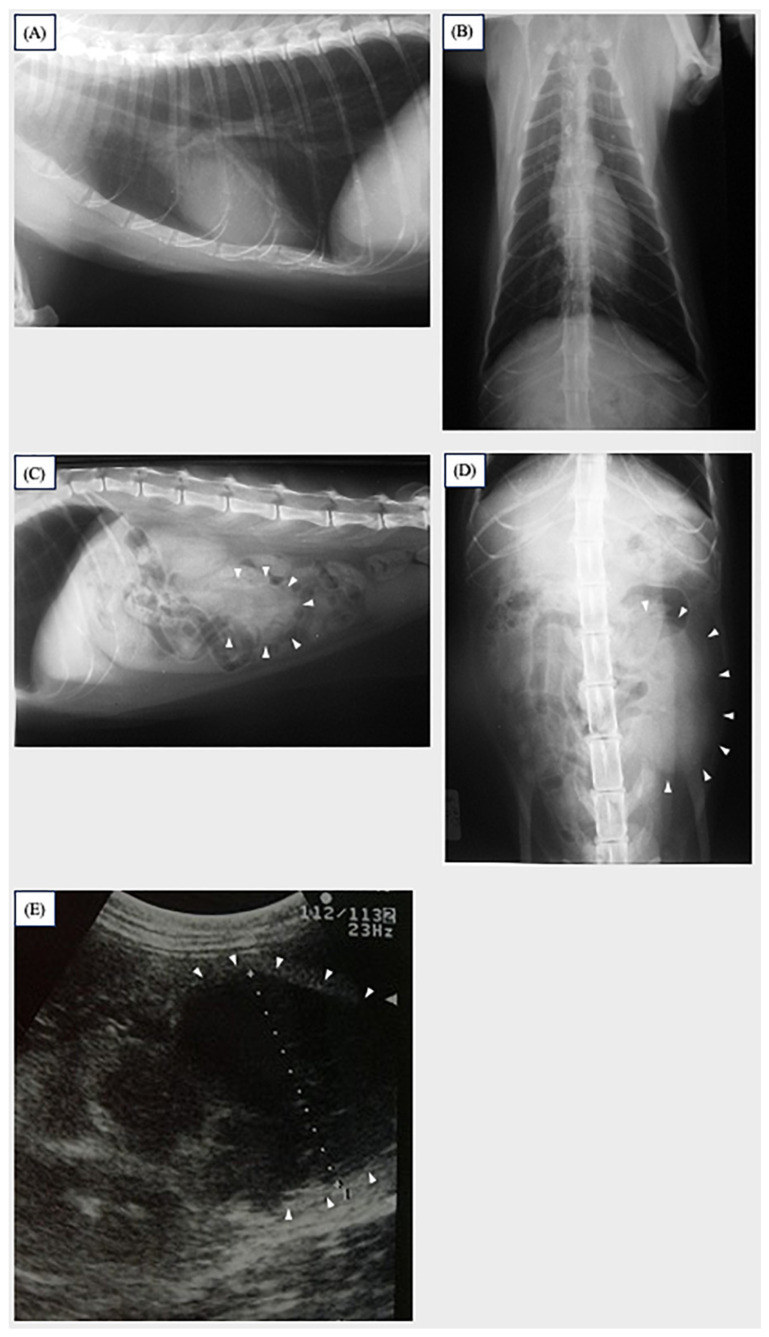
Thoracic radiographs in the right lateral (**A**) and ventral–dorsal position (**B**). No abnormal findings were observed. Abdominal radiographs in the right lateral (**C**) and ventral–dorsal position (**D**). The left kidney was enlarged and slightly displaced ventrally (white arrowhead). The ultrasound examination of the left kidney (**E**) revealed a hypoechoic renal mass located on the posterior edge of the left kidney (white arrowhead).

**Figure 3 vetsci-11-00669-f003:**
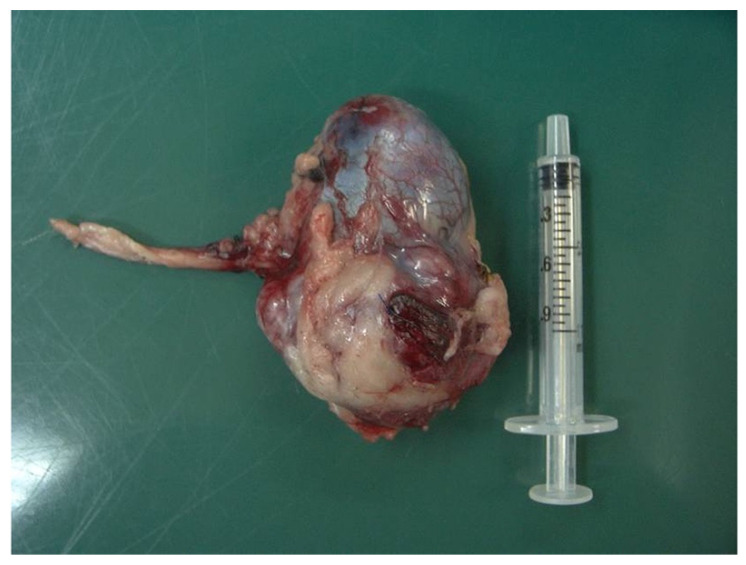
Photograph of the removed left kidney.

**Figure 4 vetsci-11-00669-f004:**
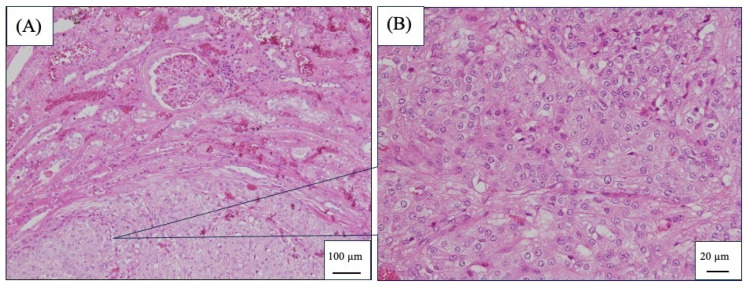
Hematoxylin- and eosin-stained slides (**A**) and magnified image (**B**) of the excised kidney.

**Figure 5 vetsci-11-00669-f005:**
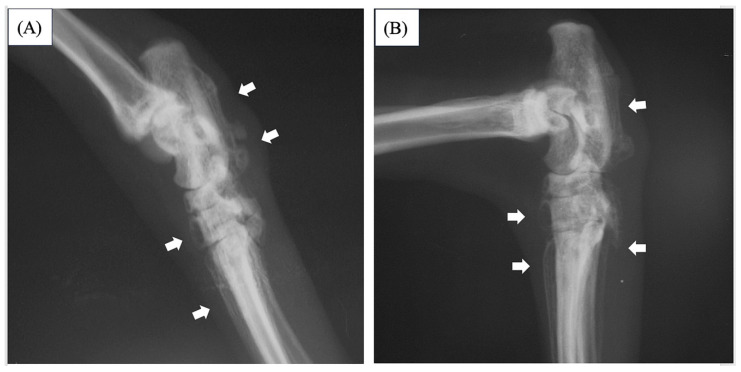
Radiographs images of a lateral view of the distal ends of the **right** (**A**) and **left** (**B**) hind limb two months after surgery. The periosteal growth of the distal tibia disappeared, whereas the periosteal proliferation of the talus and metatarsal bones decreased compared with before surgery (thick white arrow).

## Data Availability

The data presented in this study are available on request from the corresponding author.

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
