# Peer review of "Successful Treatment of Renal Cell Carcinoma Associated with Hypertrophic Osteopathy in a Cat"

_vetsci, 2024, doi:10.3390/vetsci11120669_

Round 1
Reviewer 1 Report
Comments and Suggestions for Authors
This article describes a rare case of an eight-year-old Abyssinian cat with renal carcinoma associated with hypertrophic osteopathy. The treatment consisted of a nephrectomy. Regression of the bone proliferations was observed and the cat survived for 4,500 days.
Line 3: Harmonize the font to be identical to that of line 2.
Lines 7 and 8: Provide an institutional email address if you have one.
Line 11: This is incorrect; the most common renal tumor in cats is renal lymphoma.
Line 14: Remove “similarly,” keep the uppercase on “Hypertrophic,” and replace “and” with a semicolon.
Line 15: Remove “Because the cat in the present study had developed secondary hypertrophic osteopathy to renal cell carcinoma, this represented a rare case in which nephrectomy could be performed before the clinical signs of renal cell carcinoma were evident.”
Line 18: Rephrase “This article presents a rare case of renal tumor associated with hypertrophic osteopathy that had a long survival after surgical removal of the renal tumor as well as an attenuation of bone lesions.”
Line 19: Replace “prognosis” with “survival” to indicate long-term survival, emphasizing the proven fact. Prognosis refers to the expected survival duration, which is only a hypothesis; here, survival is a proven fact.
Line 22: It is unclear which limbs are affected, and in line 23, “the affected limbs” is mentioned. It seems it may only be the hind limbs, but “stifle” is used in the singular.
Lines 24 and 26: Remove “complete blood count” and “routine serum biochemistry,” as well as line 26, which repeats the same topic. These tests have no chance of identifying the primary disease as claimed.
Line 29: The authors neither make a diagnosis nor discuss treatment. They go directly to survival!
Line 36: “Urologic neoplasias ARE uncommon” (use the plural). Do not mention domestic animals since here the discussion only concerns cats, not cattle, etc.
Line 37: Remove “other” as no other tumor has been previously mentioned!
Line 39: Rephrase to specify that lymphomas are the most common in cats, saying: “The most common renal tumor in cats is lymphoma (cite a reference); next is renal carcinoma. Renal lymphoma is almost always bilateral, so a unilateral renal tumor in a cat strongly suggests carcinoma (cite a reference).”
Line 39: Remove “conversely.”
Line 40: Replace “frequently” with “sometimes.” On the contrary, it is rare.
Line 43: Add references after “literature” to support the statement that hypertrophic osteopathy is rare in cats. Cite Johnson, R.L.; Lenz, S.D. Hypertrophic osteopathy associated with a renal adenoma in a cat. Journal of Veterinary Diagnostic Investigation 2011, 23, 171-175.
Line 44: Replace the pronoun “we” with “the present article reports.”
Line 45: Replace “prognosis” with “survival” to indicate that “long-term survival was achieved.”
Line 48: Remove the cat’s weight and replace it with “weighing 3.75 kg.”
Line 49: Delete “to our hospital.”
Line 49: Replace “she was lame” with “for lameness.” Specify the location of the lameness, presumably the hind limbs.
Line 52: It does not concern the “hind limbs” in their entirety but the distal end of the hind limbs.
Line 54: Correct figure references to a coherent nomenclature, mentioning “Figure 1A” and “Figure 1B.”
Figure 1: The images do not have the same quality (mediocre quality, in fact), so rework them with photo editing software to improve quality and make them resemble each other.
Radiograph A is bluish: convert it to black and white.
The title is incorrect. Remove “Radiographs images.” Specify that it is a lateral view of the distal ends of the right and left hind limbs.
Line 59: Remove “including mass lesions.” If there were masses, the image would therefore show anomalies.
Line 60: Refer to the examination of abdominal radiographs (in plural), face and profile, and not the “abdominal radiographic examination” in singular.
Line 60: Remove “however.”
Line 61: No, the radiograph shows a mass in the projection area of the left kidney; it is not possible at this stage to assert that it is the left kidney.
Line 62: Expand the description of the ultrasound to note that the diagnosis of renal tumor is made through this examination, and not just mention a hypoechoic mass caudal to the kidney. The statement is ambiguous: if it is caudal to the kidney, it is not the kidney.
Figure 2 - C and D:
Rename Figure 2 A, B, C, D, and E.
This is not serious work when labeled as A B and E in one row!
Remove the first sentence: Radiographs etc.
The convention is to show the lateral radiograph first and place the two thoracic radiographs side by side in the same row.
Do the same for the abdomen.
These radiographs are not homogeneous with A and B, as they have a bluish tint. Use software to harmonize in black and white, increasing sharpness and contrast.
The ultrasound shows a mass caudal to the left kidney: so it is not the left kidney? Refer to a renal mass located on the posterior edge of the left kidney (but belonging to the left kidney). At this stage, strongly suspect a renal tumor.
The images are the size of a postage stamp. Enlarge them: the journal allows this easily.
The description of the images should be placed in the preceding text.
Line 72: The author proceeds to further examinations without making any etiological hypothesis. What was the purpose of the cytological examination? There is no interpretation of the obtained result.
Line 74: State that a renal tumor is “strongly suspected,” as at this stage, there is no definitive diagnosis.
Line 75: “one week after the consultation” should be removed as it adds nothing.
Line 75: Prefer the term “nephrectomy” and omit the one-week delay.
Line 76: Avoid detailing the anesthetic protocol, which is also inappropriate, particularly due to butorphanol, which is insufficient for major surgery.
Line 78: Avoid mentioning both butorphanol and fentanyl together, as they act antagonistically. If you premedicate with butorphanol, it occupies the mu receptors without providing analgesia (it’s an antagonist), thereby blocking the fentanyl action; butorphanol is an antidote to fentanyl...
Line 80: Reference to Figure 3 should be Figure 3A, 3B, and 3C. However, photos A and B should be removed as they show nothing. Photo C should be enlarged (x2 or x3). The abdominal description should be part of the text and not in the image captions.
Figure C: “Photographs of the removed left kidney”: it is finally understood that it is the left kidney, but throughout the article, it was assumed to be a mass caudal to the left kidney.
Line 92: histological analysis… add “of the resected specimen.” The analysis description is minimal, hard to be convinced.
Figure 4: Correctly label A and B
Figure 5: Better label the figures (already mentioned). These radiographs are not homogeneous with A and B, as they have a greenish tint in radio B. Use software to harmonize in black and white, increasing sharpness and contrast.
Line 113: After the surgery (add “the”). You mention further examinations, but we don’t know how the cat is doing...
Line 119: The diagnosis of chronic renal failure is inappropriate; it would likely be acute renal failure, otherwise how to explain that it survived 3000 days with this renal failure?
Line 121: Specify if the cat was deceased or still alive at 4645 days.
Why publish this case 12 years later?
Did the periosteal lesions disappear 12 years later?
Discussion: I refrain from making a comment as detailed.
It would need to be entirely rewritten.
Add details on radiograph quality, mentioning the evolution of available image-enhancement technologies today.
Line 127: Review the discussion structure to avoid jumping from generalities to the specific case of the cat without transition, introducing for example, “In this case, the cat also presented with lameness and pain.”
Line 128: The bone biopsy was unnecessary; there is no doubt about the diagnosis of hypertrophic osteopathy, given the symmetrical and multi-bone involvement, characteristic of this pathology.
Line 151: How can the authors affirm this!!! The most common renal tumor in cats is by far lymphoma. The authors make a serious error on the central subject of their article.
Line 168: The discussion on chemotherapy is off-topic.
Conclusion
This is a beautiful case of hypertrophic osteopathy associated with a renal tumor with very long survival (we could talk about a cure),
Although the article presents some interest due to its rarity and the long survival after nephrectomy, it is regrettable that periosteal lesions were not examined after 10 years.
One drawback of this article is its age, which results in radiographic image quality being far below current standards.
The manuscript in its current form does not meet the expected quality standards for publication in Veterinary Science.
While the study addresses an interesting topic, significant improvements in both content and presentation are required.
The quality of the English needs to be improved.

The quality of English needs to be improved
Author Response
Response to Reviewer:
We wish to express our appreciation to the reviewer for their insightful comments on our paper. The comments have helped us significantly improve the paper.
Comment 1: Harmonize the font to be identical to that of line 2.
Response: We appreciate the reviewer’s comment on this point. In accordance with the reviewer’s comment, we have harmonized the font of line 3 to be identical to that of line 2.
Comment 2: Provide an institutional email address if you have one.
Response: We appreciate the reviewer’s comment on this point. In accordance with the reviewer’s comment, I have changed the institutional email address, but Takashimadaira Tezuka Animal Hospital do not have an institutional email address.
Comment 3: This is incorrect; the most common renal tumor in cats is renal lymphoma.
Response: We appreciate the reviewer’s comment on this point. We agree with the reviewer that the most common renal tumor in cats is renal lymphoma.
We have therefore added the following text (p.1, line 11)
“of which renal cell carcinoma is the most common, excluding lymphoma.”
Comment 4: Remove “similarly,” keep the uppercase on “Hypertrophic,” and replace “and” with a semicolon.
Response: We thank the reviewer for this comment. In accordance with the reviewer’s comment, we have changed the following text from (p.1, line 14)
“Similarly, Hypertrophic osteopathy in cats is rare and there are few reports in the literature.”
to
“Similarly, Hypertrophic osteopathy in cats is rare; there are few reports in the literature.”
Comment 5: Remove “Because the cat in the present study had developed secondary hypertrophic osteopathy to renal cell carcinoma, this represented a rare case in which nephrectomy could be performed before the clinical signs of renal cell carcinoma were evident.” Rephrase “This article presents a rare case of renal tumor associated with hypertrophic osteopathy that had a long survival after surgical removal of the renal tumor as well as an attenuation of bone lesions.” Replace “prognosis” with “survival” to indicate long-term survival, emphasizing the proven fact. Prognosis refers to the expected survival duration, which is only a hypothesis; here, survival is a proven fact.
Response: We thank the reviewer’s comment on this point. In accordance with the reviewer's comment, we have changed the following text from (p. 1, lines 15~20):
“Because the cat in the present study had developed secondary hypertrophic osteopathy to renal cell carcinoma, this represented a rare case in which nephrectomy could be performed before the clinical signs of renal cell carcinoma were evident. Here, we report a case of a cat with renal cell carcinoma presenting with hypertrophic osteopathy, in which a long-term prognosis was achieved following surgical treatment.”
to
“This article presents a rare case of renal tumor associated with hypertrophic osteopathy that had a long survival after surgical removal of the renal tumor as well as an attenuation of bone lesions.”
Comment 6: It is unclear which limbs are affected, and in line 23, “the affected limbs” is mentioned. It seems it may only be the hind limbs, but “stifle” is used in the singular.
Response: We thank the reviewer’s comment on this point. In accordance with the reviewer's comment, we have added the following text (p. 1, lines 22):
“Palpation revealed swelling, heat, and a reduced range of motion in the stifle and tarsal joints in the both hind limbs.”
In addition, we have changed the following text from (p. 1, lines 23)
“A radiographic examination of the affected limbs~”
to
“A radiographic examination of the both hind limbs~”
Comment 7: Remove “complete blood count” and “routine serum biochemistry,” as well as line 26, which repeats the same topic. These tests have no chance of identifying the primary disease as claimed.
Response: We thank the reviewer’s comment on this point. In accordance with the reviewer's comment, we have deleted the following text (p. 1, lines 24):
“Complete blood count, routine serum biochemistry tests, and t Thorax and abdominal radiographic~”
Comment 8: The authors neither make a diagnosis nor discuss treatment. They go directly to survival!
Response: We thank the reviewer’s comment on this point. In accordance with the reviewer's comment, we have added the following text (p. 1, lines 29):
“The left nephrectomy was performed after exploration of the abdominal cavity confirmed an abnormality in the left kidney. A histopathological examination revealed renal cell carcinoma. The cat was subsequently fed on a renal prescription diet but received no additional treatment.”
In addition, we have changed the following text from (p. 1, lines 31)
“The cat survived over 4,500 days following surgery without metastases or local recurrence.”
to
“At 1,670 days following surgery, blood tests indicated that urea nitrogen and creatinine were above the normal range. However, the cat survived over 4,500 days following surgery.”
Comment 9: “Urologic neoplasias ARE uncommon” (use the plural). Do not mention domestic animals since here the discussion only concerns cats, not cattle, etc.
Response: We thank the reviewer’s comment on this point. In accordance with the reviewer's comment, we have deleted the following text (p. 1, lines 35-37):
Urologic neoplasia, including primary renal tumors and tumors of the bladder or urethra, is uncommon in domestic animals, accounting for less than 1% of all neoplastic conditions and 2% of all malignant tumors [1].
Comment 10: Remove “other” as no other tumor has been previously mentioned!
Rephrase to specify that lymphomas are the most common in cats, saying: “The most common renal tumor in cats is lymphoma (cite a reference); next is renal carcinoma. Renal lymphoma is almost always bilateral, so a unilateral renal tumor in a cat strongly suggests carcinoma (cite a reference)
Response: We thank the reviewer’s comment on this point. In accordance with the reviewer's comment, we have changed the following text from (p. 1, lines 37-39):
“Reports of other primary feline renal tumors are rare and generally limited to case reports and surveys of tumors, of which renal cell carcinoma is the most common, except lymphoma.”
to
“Reports of primary feline renal tumors are rare and generally limited to case reports and surveys of tumors. Most common renal tumor in cats is lymphoma; next is renal carcinoma. Renal lymphoma is almost always bilateral, so a unilateral renal tumor in a cat strongly suggests carcinoma.”
Comment 11: Remove “conversely.” Replace “frequently” with “sometimes.” On the contrary, it is rare.”
Response: We thank the reviewer’s comment on this point. In accordance with the reviewer's comment, we have changed the following text from (p. 1, lines 39-40):
“Conversely, hypertrophic osteopathy (HO) is frequently described in~”
to
“Conversely, Hypertrophic osteopathy (HO) is sometimes described in~”
Comment 12: Add references after “literature” to support the statement that hypertrophic osteopathy is rare in cats. Cite Johnson, R.L.; Lenz, S.D. Hypertrophic osteopathy associated with a renal adenoma in a cat. Journal of Veterinary Diagnostic Investigation 2011, 23, 171-175.
Response: We thank the reviewer’s comment on this point. In accordance with the reviewer's comment, we have added the references (p. 1, lines 39-40):
“HO in cats is rare and there are few reports in the literature [Johnson, R.L.; Lenz, S.D. Hypertrophic osteopathy associated with a renal adenoma in a cat. Journal of Veterinary Diagnostic Investigation 2011, 23, 171-175.].”
Comment 13: Replace the pronoun “we” with “the present article reports.” Replace “prognosis” with “survival” to indicate that “long-term survival was achieved.”
Response: We thank the reviewer’s comment on this point. In accordance with the reviewer's comment, we have changed the following text from (p. 1, lines 44):
“Here, we report a case of a cat with renal cell carcinoma presenting with HO, in which a long-term prognosis was achieved following surgical treatment.”
to
“Here, the present article reports a case of a cat with renal cell carcinoma presenting with HO, in which a long-term survival was achieved following surgical treatment.”
Comment 14: Remove the cat’s weight and replace it with “weighing 3.75 kg.” Delete “to our hospital.” Replace “she was lame” with “for lameness.” Specify the location of the lameness, presumably the hind limbs.
Response: We thank the reviewer’s comment on this point. In accordance with the reviewer's comment, we have changed the following text from (p. 2, lines 48):
“An eight-year-old spayed female Abyssinian cat (body weight: 3.75 kg) presented to our hospital because she was lame for several months, unable to bend her limbs or climb to high places.”
to
“An eight-year-old spayed female Abyssinian cat (weighting 3.75 kg) presented to our hospital because she was lameness for several months, unable to bend her hind limbs or climb to high places.”
Comment 15: It does not concern the “hind limbs” in their entirety but the distal end of the hind limbs.
Response: We thank the reviewer’s comment on this point. In accordance with the reviewer's comment, we have added the following text (p. 2, lines 52):
“A radiographic examination of both hind limbs revealed soft tissue swelling the distal end of the hind limbs and~
”
Comment 16: Correct figure references to a coherent nomenclature, mentioning “Figure 1A” and “Figure 1B.” Correct figure references to a coherent nomenclature, mentioning “Figure 1A” and “Figure 1B.” The images do not have the same quality (mediocre quality, in fact), so rework them with photo editing software to improve quality and make them resemble each other.
Radiograph A is bluish: convert it to black and white. The title is incorrect. Remove “Radiographs images.” Specify that it is a lateral view of the distal ends of the right and left hind limbs.
Response: We appreciate the reviewer’s comment on this point. In accordance with the reviewer's comment, we have edited the images to improve quality for Figure 1 and convert it to black and white. In addition, we have changed the following Figure Legends (p. 2, lines 56)
“Radiographs images of a lateral view of the distal ends of the right and left hind limbs. Radiographic examination of the right hind limb (A) and left hind limb (B)~”
Comment:17 Remove “including mass lesions.” If there were masses, the image would therefore show anomalies. Refer to the examination of abdominal radiographs (in plural), face and profile, and not the “abdominal radiographic examination” in singular. Remove “however.” No, the radiograph shows a mass in the projection area of the left kidney; it is not possible at this stage to assert that it is the left kidney.
Response: We thank the reviewer’s comment on this point. In accordance with the reviewer's comment, we have changed the following text from (p. 2, lines 59-61):
“A thoracic radiographic examination showed no abnormal findings, including mass lesions (Figure 2A, B); however, an abdominal radiographic examination showed enlargement of the left kidney (Figure 2C, D).”
to
“The examination of thoracic radiographs showed no abnormal findings, including mass lesions (Figure 2A, B). however, The examination of abdominal radiographs showed a mass in the projection area of the left kidney (Figure 2C, D).”
Comment 18: Expand the description of the ultrasound to note that the diagnosis of renal tumor is made through this examination, and not just mention a hypoechoic mass caudal to the kidney. The statement is ambiguous: if it is caudal to the kidney, it is not the kidney.
Response: We thank the reviewer’s comment on this point. We agree with the reviewer that if it is caudal to the kidney, it is not the kidney. To make this point clearer, we have changed the following text from (p. 2, lines 59-61):
“Abdominal ultrasound revealed a hypoechoic mass caudal to the left kidney (Figure 2E).”
to
“Abdominal ultrasound revealed a hypoechoic renal mass located on the posterior edge of the left kidney (Figure 2E). The mass was 27 mm and exhibited a clear border with the surrounding tissue.”
Comment 19: Rename Figure 2 A, B, C, D, and E. This is not serious work when labeled as A B and E in one row! Remove the first sentence: Radiographs etc. The convention is to show the lateral radiograph first and place the two thoracic radiographs side by side in the same row.
Do the same for the abdomen. These radiographs are not homogeneous with A and B, as they have a bluish tint. Use software to harmonize in black and white, increasing sharpness and contrast. The ultrasound shows a mass caudal to the left kidney: so it is not the left kidney? Refer to a renal mass located on the posterior edge of the left kidney (but belonging to the left kidney). At this stage, strongly suspect a renal tumor. The images are the size of a postage stamp. Enlarge them: the journal allows this easily. The description of the images should be placed in the preceding text.
Response: We appreciate the reviewer’s comment on this point. In accordance with the reviewer’s comment, we have changed the figure 2 and figure legends.
“Thoracic radiographs in the right lateral (A) and ventral-dorsal position (B). No abnormal findings were observed. Abdominal radiographs in the right lateral (C) and ventral-dorsal position (D). The left kidney was enlarged and slightly displaced ventrally (white arrowhead). Ultrasound examination of the left kidney (E) revealed a hypoechoic renal mass located on the posterior edge of the left kidney (white arrowhead).”
Comment 20: The author proceeds to further examinations without making any etiological hypothesis. What was the purpose of the cytological examination? There is no interpretation of the obtained result.
Response: We thank the reviewer’s comment on this point. In accordance with the reviewer's comment, we have changed the following text from (p. 3, lines 72):
“Subsequently, fine needle aspiration was performed on~”
to
“For the purpose of diagnosing the left renal mass, fine needle aspiration was performed on~”
Comment 21: State that a renal tumor is “strongly suspected,” as at this stage, there is no definitive diagnosis.
Response: We thank the reviewer’s comment on this point. In accordance with the reviewer's comment, we have changed the following text from (p. 3, lines 74):
“the cat was diagnosed with a left renal tumor associated with HO.”
to
“the cat was strongly suspected with a left renal tumor associated with HO.”
Comment 22: “one week after the consultation” should be removed as it adds nothing. Prefer the term “nephrectomy” and omit the one-week delay.
Response: We thank the reviewer’s comment on this point. In accordance with the reviewer's comment, we have changed the following text from (p. 3, lines 75~76):
“Surgical excision of the left renal tumor was performed one week after the consultation.”
to
“Nephrectomy of the left renal tumor was performed one week after the consultation.”
Comment 23: Avoid detailing the anesthetic protocol, which is also inappropriate, particularly due to butorphanol, which is insufficient for major surgery. Avoid mentioning both butorphanol and fentanyl together, as they act antagonistically. If you premedicate with butorphanol, it occupies the mu receptors without providing analgesia (it’s an antagonist), thereby blocking the fentanyl action; butorphanol is an antidote to fentanyl...
Response: We thank the reviewer’s comment on this point. We agree with the reviewer that butorphanol is insufficient and act antagonistically to fentanyl. We did not use butorphanol as a premedication and this was an error in our description. we have therefore changed the following text from (p. 3, lines 76-77):
“Premedication was administered with midazolam (0.2 mg/kg IV) and butorphanol (0.2 mg/kg IV).”
to
“Premedication was administered with fentanyl (5 μg/kg IV).”
Comment 24: Reference to Figure 3 should be Figure 3A, 3B, and 3C. However, photos A and B should be removed as they show nothing. Photo C should be enlarged (x2 or x3). The abdominal description should be part of the text and not in the image captions. “Photographs of the removed left kidney”: it is finally understood that it is the left kidney, but throughout the article, it was assumed to be a mass caudal to the left kidney.
Response: We thank the reviewer’s comment on this point. We agree with the reviewer that
photos A and B should be removed as they show nothing. However, we decided that it would be easier to understand the circumstances of the surgery if the photograph showing the abdominal cavity before the nephrectomy. we have therefore deleted the Figure (B) and Figure (C) was enlarged. In addition, we have changed figure legends.
Figure 3. Photograph during surgery. The top of the photograph is the cranial direction and the bottom is the caudal direction (A). Photograph of the removed left kidney (B).
In addition, we have changed the following text from (p. 3, lines 88-89):
“Exploration of the abdominal cavity confirmed a voluminous mass deforming the left kidney.”
to
“Exploration of the abdominal cavity indicated a clear serous fluid accumulated under the retroperitoneum. The retro-peritoneum were incised to expose the left kidney, voluminous mass deforming the left kidney was confirmed.”
comment 25: histological analysis… add “of the resected specimen.” The analysis description is minimal, hard to be convinced. Correctly label A and B
Response: We thank the reviewer’s comment on this point. In accordance with the reviewer's comment, we have added the following text (p. 3, lines 93)
“The tumor cells were irregularly round glandular epithelial cells and formed a solid glandular tubular structure. They were formed protruding outward from the renal cortex. There was a slight variation in size among the tumor cells and a small number of mitotic figures were observed in the irregularly round atypical nuclei.”
In addition, we have changed the figure legends from (p.4, lines 99~101):
“Hematoxylin- and eosin-stained photograph (A) and magnified image (B) of the excised kidney. The tumor cells were irregularly round glandular epithelial cells and formed a solid glandular tubular structure. There was a slight variation in size among the tumor cells and mitotic figures were observed in the irregularly round atypical nuclei.”
to
“Hematoxylin- and eosin-stained slides (A) and magnified image (B) of the excised kidney.”
comment 26: Better label the figures (already mentioned). These radiographs are not homogeneous with A and B, as they have a greenish tint in radio B. Use software to harmonize in black and white, increasing sharpness and contrast.
Response: We appreciate the reviewer’s comment on this point. The radiographs images were film-based and the conditions could not be changed, there was a limit to improvement. Also, to provide a more easily viewable image, the lateral view of the distal ends of the right hind limbs in Figure 5 was taken on the same day, but it has been changed to a different image. In accordance with the reviewer's comment, we have edited the images to improve quality for Figure 5 and convert it to black and white. In addition, we have changed the following Figure Legends (p. 5, lines 110)
“Figure 5. Radiographs images of a lateral view of the distal ends of the right (A) and left (B) hind limb two months after surgery.
comment 2ï¼—: After the surgery (add “the”). You mention further examinations, but we don’t know how the cat is doing...
Response: We thank the reviewer’s comment on this point. The cat presented because she was lameness. Therefore, we only describe the clinical symptoms of lameness. We scripted these things from lines 104 on page 4. In addition, in accordance with the reviewer's comment, we have added the following text (p. 4, lines 103)
“one month after the surgery”
Comment 28: The diagnosis of chronic renal failure is inappropriate; it would likely be acute renal failure,
Response: We thank the reviewer’s comment on this point. Clinically, chronic renal failure (CKD) in cats usually is diagnosed based on a combination of polyuria, polydipsia, azotemia, and an inappropriate urine specific gravity (Sparkes AH, Caney S, Chalhoub S, et al. ISFM consensus guidelines on the diagnosis and management of feline chronic kidney disease. J Feline Med Surg. 2016; 18: 219-239.). The clinical presentation of acute kidney injury (AKI) cats, as in conditions the most common lethargy, anorexia, and vomiting (Segev G, Nivy R, Kass PH, Cowgill LD. A retrospective study of acute kidney injury in cats and development of a novel clinical scoring system for predicting outcome for cats managed by hemodialysis. J Vet Intern Med. 2013; 27: 830-839.). Since AKI and CKD in cats and dogs are interconnected processes recently has been proposed, it is very difficult to make this distinction. As the reviewer comment, acute kidney injury was a possibility at that time, but based on the persistent azotemia in subsequent blood examinations and no abnormalities could be detected on the abdominal ultrasound, we suspected the cat with chronic kidney disease.
to make this point clear, we have changed the following text from (p. 5, lines 117):
“Although an abdominal ultrasound showed no abnormalities in the right kidney, the patient was diagnosed with chronic kidney disease because of continued low specific gravity urine (1.015) and azotemia.”
to
“An abdominal ultrasound showed no abnormalities in the right kidney. The cat was suspected with chronic kidney disease because of continued azotemia and low specific gravity urine (1.015).”
Comment 29: otherwise how to explain that it survived 3000 days with this renal failure?
Response: We acknowledge the reviewer’s comment on this point. We agree with your comment that the time of death is much longer than the average life span for chronic renal disease (CKD). Median survival time for CKD cats was about 1 year, which is similar to other reports in the literature. Conversely, CKD is a chronic progressive disease, individual survival times vary: some cases have been reported with survival times exceeding 2000 days, but some cats have lived longer because they were unable to be followed up or survived the study period. The cat's owner continued to visit the hospital until the cat's death, so we were able to follow its progress until the end.
Dietary therapy is very important in managing CKD in cats. According to the International Renal Interest Society (IRIS) guidelines, renal prescription diets are recommended from stage 2 onwards. However, the usefulness of therapeutic renal diets has also been reported for cats with non-proteinuria IRIS CKD stage 1. Therefore, this cat's longevity may have been due to one of the reasons it was able to feed its renal prescription diet earlier.
To make this point clearer, we have added the following text from (p. 4, lines 108):
“A renal prescription diet was started in consideration of the remaining right kidney at this point.”
Comment 30: Specify if the cat was deceased or still alive at 4645 days. Why publish this case 12 years later? Did the periosteal lesions disappear 12 years later?
Response: We acknowledge the reviewer’s comment on this point. The cat's last blood test performed 3.5 years after the date of CKD diagnosis, showed that urea nitrogen and creatinine were 40.7 mg/dL and 2.4 mg/dL, respectively. Neither anemia nor proteinuria was confirmed at that time. Since the owner did not request for any further tests, no tests were performed. The cat's weight and appetite were gradually reduced in the final stages, so one possible cause of death was the worsening of CKD, but since a necropsy was not performed, it was difficult to determine the cause of death. The 12 years is simply a result of following this cat's progress until her death, and we never expected her to live more than 10 years when we began treatment. About Affected hind limbs, six months after surgery, the radiographic examinations showed that these lesions did not decrease. Since lameness symptoms had improved, we have not performed follow-up the radiographic examination of both hind limbs.
Therefore, we have changed the following text from (p. 5, lines 120-121):
“However, no metastasis or local recurrence of renal cell carcinoma was observed and the cat survived 4645 days following the operation.”
to
“The cat's last blood test performed eight years after the surgery, showed that urea nitrogen and creatinine were 40.7 mg/dL and 2.4 mg/dL, respectively. Anemia or proteinuria was not noted at the time. The cat was subsequently fed on a renal prescription diet but received no additional examinations and treatments. The cat's vitality and appetite gradually decreased and died 4645 days following the operation. The cause of death could not be determined as no autopsy was performed.
In addition, we have added the following text (p. 4, lines 108)
“Six months after surgery, the radiographic examinations showed that these lesions had not decreased, but there were no clinical symptoms such as lameness, follow-up radiographic examinations of both hind limbs were not performed.”
Comment 31: Review the discussion structure to avoid jumping from generalities to the specific case of the cat without transition, introducing for example, “In this case, the cat also presented with lameness and pain.”
Response: We thank the reviewer’s comment on this point. In accordance with the reviewer's comment, we have changed the following text from (p. 5, lines 127):
“The cat also presented with HO, but also showed lameness and pain.”
to
“In this case, the cat also presented with lameness and pain.”
Comment 31: The bone biopsy was unnecessary; there is no doubt about the diagnosis of hypertrophic osteopathy, given the symmetrical and multi-bone involvement, characteristic of this pathology.
Response: We thank the reviewer’s comment on this point. We agree with your comment that here is no doubt about the diagnosis of hypertrophic osteopathy. But, to make a definitive diagnosis, not only imaging findings but also cytological diagnosis is required.
Comment 31: How can the authors affirm this!!! The most common renal tumor in cats is by far lymphoma. The authors make a serious error on the central subject of their article.
Response: We thank the reviewer’s comment on this point. In accordance with the reviewer's comment, we have changed the following text from (p. 6, lines 149-151):
“The majority of tumors are epithelial, with renal cell carcinoma considered the most common~”
to
“Except lymphoma, the majority of tumors are epithelial, with renal cell carcinoma considered the most common~”
Comment 32: The discussion on chemotherapy is off-topic.
Response: We thank the reviewer’s comment on this point. Regardless of tumor type, primary renal tumors have a high risk of metastasis and some form of adjuvant chemotherapy is usually recommended. In this article, adhesions were observed between the caudal side of the left kidney and the fat in the retroperitoneum. Therefore, we thought it was also necessary to discuss chemotherapy.
This article's radiographs from over 10 years ago were on films and are obviously inferior in quality to today's digital images. I have processed them as much as possible to make them easier for readers to view, but I think it is important to be careful not to edit them too much.
The paper has been edited and rewritten by an experienced scientific editor, who has improved the grammar and stylistic expression the paper. In addition, in accordance with the reviewer's comment, we have significantly improved the text.
Thank you again for your comments on your paper. We trust that the revised manuscript is suitable for publication.

Reviewer 2 Report
Comments and Suggestions for Authors
Overall, this manuscript presents an interesting case report of feline hypertrophic osteopathy (HO) associated with renal cell carcinoma. The unique nature of the case report is the outcome of prolonged survival after surgical removal of the tumor-involved kidney associated with regression of HO, although regression was only partial. The report is relatively well-written and adheres to the required format of a case report for this journal. However, there is one concern that should be addressed by the authors.
Comment:
1. Please confirm that time frames of 1,670 days post surgery as the time of diagnosis of chronic renal disease and survival to 4,465 days post surgery. Given this cat was 8 years old at the time of surgery, she was approximately 20 years of age at death, a lifespan that is reported for domestic cats but hovers at the maximum age achieved by cats. However, the time frame of 8 years (4,645 - 1,670 days) between the diagnosis of chronic renal disease and time of death is much longer than the average life span for this diagnosis as reported in the literature. Please address this concern and report the cause of death if known.
Author Response
Response to Reviewer:
We wish to express our appreciation to the associate statistical editor for their insightful comments on our paper. The comments have helped us significantly improve the paper.
Comment 1: Please confirm that time frames of 1,670 days post surgery as the time of diagnosis of chronic renal disease and survival to 4,465 days post surgery. Given this cat was 8 years old at the time of surgery, she was approximately 20 years of age at death, a lifespan that is reported for domestic cats but hovers at the maximum age achieved by cats. However, the time frame of 8 years (4,645 - 1,670 days) between the diagnosis of chronic renal disease and time of death is much longer than the average life span for this diagnosis as reported in the literature. Please address this concern and report the cause of death if known.
Response: We acknowledge the reviewer’s comment on this point. We agree with your comment that the time of death is much longer than the average life span for chronic renal disease (CKD). Median survival time for CKD cats was about 1 year, which is similar to other reports in the literature. Conversely, CKD is a chronic progressive disease, individual survival times vary: some cases have been reported with survival times exceeding 2000 days, but some cats have lived longer because they were unable to be followed up or survived the study period. In this article, the cat's owner continued to visit the hospital until the cat's death, so we were able to follow its progress until the end.
Dietary therapy is very important in managing CKD in cats. According to the International Renal Interest Society (IRIS) guidelines, renal prescription diets are recommended from stage 2 onwards. However, the usefulness of therapeutic renal diets has also been reported for cats with non-proteinuria IRIS CKD stage 1. Therefore, this cat's longevity may have been due to one of the reasons it was able to feed its renal prescription diet earlier. The cat's last blood test performed 3.5 years after the date of CKD diagnosis, showed that urea nitrogen and creatinine were 40.7 mg/dL and 2.4 mg/dL, respectively. Neither anemia nor proteinuria was confirmed at that time. Since the owner did not request for any further tests, no tests were performed. The cat's weight and appetite were gradually reduced in the final stages, so one possible cause of death was the worsening of CKD, but since a necropsy was not performed, it was difficult to determine the cause of death.
To make this point clearer, we have added the following text from (p. 4, lines 108):
“A renal prescription diet was started in consideration of the remaining right kidney at this point.”
In addition, we have changed the following text from (p. 5, lines 117):
“Although an abdominal ultrasound showed no abnormalities in the right kidney, the patient was diagnosed with chronic kidney disease because of continued low specific gravity urine (1.015) and azotemia.”
to
“An abdominal ultrasound showed no abnormalities in the right kidney. The cat was suspected with chronic kidney disease because of continued azotemia and low specific gravity urine (1.015).”
In addition, we have changed the following text from (p. 5, lines 120-121):
“However, no metastasis or local recurrence of renal cell carcinoma was observed and the cat survived 4645 days following the operation.”
to
“The cat's last blood test performed eight years after the surgery, showed that urea nitrogen and creatinine were 40.7 mg/dL and 2.4 mg/dL, respectively. Anemia or proteinuria was not noted at the time. The cat was subsequently fed on a renal prescription diet but received no additional examinations and treatments. The cat's vitality and appetite gradually decreased and died 4645 days following the operation. The cause of death could not be determined as no autopsy was performed.
Thank you again for your comments on your paper. We trust that the revised manuscript is suitable for publication.

Reviewer 3 Report
Comments and Suggestions for Authors
Dear Author,
Your manuscript, "Successful Treatment of Renal Cell Carcinoma Associated with Hypertrophic Osteopathy in a Cat," is well-written. You used all parts of scientific articles, including the case report (introduction, case description, discussion, and conclusion). The list of references is ok. In line 98 - in the description of Figure 4, you make a mistake. Hematoxylin and eosin are stained slides, not microphotographs. Please change. How big was the mitotic index of the neoplastic cells? Was this tumor highly malignant? Was it localized in the renal cortex or medulla? Please explain.
Best regards
Author Response
Response to Reviewer:
We wish to express our appreciation to the reviewer for their insightful comments on our paper.
Comment 1: Your manuscript, "Successful Treatment of Renal Cell Carcinoma Associated with Hypertrophic Osteopathy in a Cat," is well-written. You used all parts of scientific articles, including the case report (introduction, case description, discussion, and conclusion). The list of references is ok. In line 98 - in the description of Figure 4, you make a mistake. Hematoxylin and eosin are stained slides, not microphotographs. Please change. How big was the mitotic index of the neoplastic cells? Was this tumor highly malignant? Was it localized in the renal cortex or medulla? Please explain.
Response: We appreciate the reviewer’s comment on this point. In accordance with the reviewer's comment, we have changed the following text from (p. 4, lines 98):
“Hematoxylin- and eosin-stained photograph (A) and magnified image (B) of the excised kidney. The tumor cells were irregularly round glandular epithelial cells and formed a solid glandular tubular structure. There was a slight variation in size among the tumor cells and mitotic figures were observed in the irregularly round atypical nuclei.”
to
“Hematoxylin- and eosin-stained slides (A) and magnified image (B) of the excised kidney.”
In addition, we have added the following the text (p.3, lines 93):
“The tumor cells were irregularly round glandular epithelial cells and formed a solid glandular tubular structure. They were formed protruding outward from the renal cortex. There was a slight variation in size among the tumor cells and a small number of mitotic figures were observed in the irregularly round atypical nuclei.”
Thank you again for your comments on your paper. We trust that the revised manuscript is suitable for publication.
